# The Role of Intravesicular Proteins and the Protein Corona of Extracellular Vesicles in the Development of Drug-Induced Polyneuropathy

Natalia V. Yunusova [1,2,*], Natalia O. Popova [3], Irina N. Udintseva [3], Tatyana S. Klyushina [2], Daria V. Kazantseva [4] and Liudmila P. Smirnova [4]

1    Laboratory of Tumor Biochemistry, Cancer Research Institute, Tomsk National Research Medical Center, Tomsk 634009, Russia
2    Department of Biochemistry and Molecular Biology, Siberian State Medical University, Tomsk 634050, Russia; klyushina.tatyana@mail.ru
3    Department of Chemotherapy, Cancer Research Institute, Tomsk National Research Medical Center, Tomsk 634009, Russia; popova75tomsk@mail.ru (N.O.P.); nusya131@yandex.ru (I.N.U.)
4    Laboratory of Molecular Genetics and Biochemistry, Mental Health Research Institute, Tomsk National Research Medical Center, Tomsk 634009, Russia; dashka1745@mail.ru (D.V.K.); lpsmirnova2016@gmail.com (L.P.S.)
*    Correspondence: bochkarevanv@oncology.tomsk.ru; Tel.: +7-9234239879

**Abstract:** Extracellular vesicles (EVs) as membrane structures of cellular origin participating in intercellular communication are involved in the molecular mechanisms of the development of various variants of polyneuropathy. Taking into account the increasing role of the protein corona of EVs and protein-protein interactions on the surface of EVs in the pathogenesis of various diseases, we focused our attention in this review on the role of intravesicular proteins and the protein corona of EVs in the development of chemotherapy-induced polyneuropathy (CIPN). It has been shown that EVs are effectively internalized by the mechanisms of endocytosis and macropinocytosis by neurocytes and glial cells, carry markers of insulin resistance, functionally active proteins (receptors, cytokines, enzymes), and may be involved in the pathogenesis of CIPN. The mechanisms of CIPN associated with the EVs protein corona can be related with the accumulation of heavy chains of circulating IgG in it. G-class immunoglobulins in EVs are likely to have myelin hydrolyzing, superoxide dismutase, and oxidoreductase enzymatic activities. Moreover, circulating IgG-loaded EVs are a place for complement activation that can lead to membrane attack complex deposition in neuroglia and neurons. The mechanisms of CIPN development that are not associated with IgG in the EVs protein corona are somehow related to the fact that many anticancer drugs induce apoptosis of tumor cells, neurons, and neuroglial cells by various mechanisms. This process may be accompanied by the secretion of EVs with modified cargo (HSPs, 20S proteasomes, miRNAs).

**Keywords:** extracellular vesicles; protein corona; chemotherapy-induced polyneuropathy; anticancer drug; oxaliplatin

Polyneuropathy is a diffuse disorder of the peripheral nerves that is not limited to a single nerve or a single limb but is usually relatively symmetrical on both sides. Usually infections, toxins, drug, cancers, diabetes mellitus, autoimmune disorders can cause many peripheral nerves to malfunction. Extracellular vesicles (EVs) as membrane structures of cellular origin participating in intercellular communication are involved in the molecular mechanisms of the development of various variants of polyneuropathy.

## 1. Proteins and miRNAs of Extracellular Vesicles (EVs) as Biomarkers of Diabetic Polyneuropathy

It has been shown that EVs are effectively internalized by the endocytosis and macropinocytosis by neurocytes and glial cells, carry markers of insulin resistance, functionally

active proteins (receptors, cytokines, enzymes), and may be involved, for example, in the pathogenesis of diabetic polyneuropathy [1,2]. It is known that the vascular endothelium is the primary target of insulin. As promising vesicular markers associated with endothelial dysfunction in Type 2 diabetes mellitus (DM2) are the level of CD31+CD144+ EVs, the level of miR-126 in circulating EVs, and an increased ratio of CD31+Annexin V+ EVs to CD31+CD62+ EVs, reflecting the altered immune phenotype of endothelial cell-derived EVs [3,4]. Vesicular markers are successfully used as markers of an increased risk of cardiovascular disease complications in obesity, DM2, and metabolic syndrome (perilipin A, cystatin C, CD14, miR21, disulfide isomerase levels on CD31-positive BBs) [3,5–7]. Several miRNAs have been identified in urinary EVs associated with diabetic nephropathy and renal fibrosis. In addition, exosomal miRNAs have been identified in circulating exosomes associated with albuminuria in patients with diabetic nephropathy [8–10].

It was found that a number of miRNAs secreted as part of EVs are involved in the pathogenesis of diabetic retinopathy and neuropathy, and in the future, they can also be useful as early and reliable markers of these complications [11–13].

Mesenchymal stem cell-derived EVs are also discussed as promising products for regenerative medicine, and in particular for the treatment of diabetic polyneuropathy [14–16].

## 2. Chemotherapy-Induced Polyneuropathy as a Clinical Challenge

Chemotherapy (CT) is one of the most common treatment options for cancer. However, severe side effects of CT often lead to the need for dosage adjustment, delay insubsequent cycles of CT and sometimes to complete cessation of treatment. Neurotoxicity is one of the specific complications of CT [17]. Peripheral polyneuropathy is the most common manifestation of neurotoxicity resulting from damage to peripheral motor, sensory and autonomic neurons. In comparison to other peripheral neuropaties, chemotherapy-induced polyneuropathy (CIPN) is characterized by the predominant involvement of sensory and motor fibers, dependence on the dose and type of chemotherapeutic agents, and symptom relief after the withdrawal of the agents that caused CIPN [18,19]. Chemotherapeutic agents most commonly associated with CIPN are shown in Table 1 [20–24].

**Table 1.** Anticancer drugs as a cause CIPN.

| Substance Group | Chemotherapy Drugs | Drug Indications | References |
|---|---|---|---|
| Vinca alkaloids | Vincristine Vinblastine Vinorelbine | Acute leukemias, lymphogranulomatosis, non-Hodgkin's lymphomas, multiple myeloma, Ewing's sarcoma, osteogenic sarcoma, soft tissue sarcomas, breast cancer, small cell lung cancer, melanoma, bladder cancer, Wilms tumor, neuroblastoma, cervical cancer, uterine sarcoma | [23,24] |
| Platinum-based drugs | Cisplatin Carboplatin Oxaliplatin | Small and non-small cell lung cancer; squamous cell carcinoma of various localizations (brain, stomach, esophagus, bladder, cervix); ovarian cancer; osteosarcoma, colorectal cancer | [19,21] |
| Proteasome inhibitors | Bortezomib | Multiple myeloma | [17,18] |
| Taxanes | Paclitaxel, docetaxel | Breast cancer, ovarian cancer, gastric cancer | [20–24] |
| Non-taxane microtubule dynamics inhibitor | Eribulin | Breast cancer, liposarcomas, leiomyosarcomas | [24] |

Widely used in the treatment of gastric and colorectal cancer, oxaliplatin is included in modern chemotherapy regimens FOLFOX, XELOX, FLOX, FLOT. Taxanes have been successfully used in second-line chemotherapy in patients with gastric cancer after tumor progression on first-line chemotherapy.

Clinically, CIPN is manifested by a complex of motor, sensory and autonomic disorders, which can be negative (symptoms of prolapse) or positive (symptoms of irritation) depending on the depth of the corresponding fiber type. Symptoms may be isolated, but more often they are in combination and have varying degrees of severity. Positive sensory symptoms (burning sensation and other paresthesias) usually appear first in the feet, most commonly in the fingertips. Negative sensory symptoms (numbness and decreased sensitivity) then appear and gradually spread in the proximal direction, as shorter fibers are affected [22,23]. Superficial sensory disturbances in the hands usually appear only after the symptoms in the legs have risen to the middle of the legs, which leads to the appearance of the classic "socks and gloves". Positive sensory disorders also include pain, dysesthesia, hyperesthesia, hyperpathia, and hyperalgesia. Movement disorders in CIPN typically present with flaccid tetraparesis that initially involves the muscles of the feet and lower legs and then spreads proximally. The characteristic features include a slowly progressive increase in symptoms; symmetrical distal involvement of the hands and feet; symptoms of peripheral neurosensory dysfunction (paresthesia, dysesthesia, hypesthesia, deterioration of proprioceptive, vibrational, temperature, tactile sensitivity). In advanced cases, a significant limitation of the patient's daily activity develops (the impossibility of writing, walking, using a computer keyboard, cutlery, etc.). A concomitant impairment of motor function occurs rather late (mild or moderate muscle weakness, muscle atrophy) [22,23]. Polyneuropathies induced by a specific drug in some cases have their own characteristics. Thus, vincristine-associated polyneuropathy is characterized by constipation, episodes of episodic weakness, gait disturbances, erectile dysfunction, orthostatic hypotension, while weakness of the distal limb muscles, neuropathic pain, and vegetative disorders are uncharacteristic. Neuropathic pain caused by damage to the somatosensory nervous system is a specific feature of polyneuropathy induced by proteasome inhibitors and immunomodulatory agents [19–21]. Severity of CIPN is directly dependent on the cumulative dose of the chemotherapy drug and the treatment regimen used, and the simultaneous combination of several neurotoxic cytostatics aggravates its course. The risk of damage to the peripheral nervous system is increased in cancer patients with hereditary and acquired neuropathies; suffering from diabetes; alcohol abusers; having hepatic and/or renal dysfunction; previously treated with any neurotoxic drugs (especially vinca alkaloids, taxanes, platinum derivatives, methotrexate) [24].

## 3. Pathogenesis of Polyneuropathy Caused by the Use of Anticancer Drugs

Unlike the central nervous system, peripheral axons are not protected by the blood-brain barrier, which allows metabolites of cytostatics to penetrate into the nerve fibers from the surrounding interstitial fluid by direct diffusion and accumulate in them, causing their damage. Most likely, most of the manifestations of polyneuropathy are the result of a violation of the microtubular architectonics of axons along with direct damage to the distal axons (axonopathy), diffuse or segmental demyelination of neurons (myelinopathy) or degeneration of their bodies (neuronopathy) [25]. Several molecular mechanisms that determine the development of CIPN, which often complement each other, can be identified [26,27]:

- disruption of microtubule dynamics and axon transport;
- oxidative stress;
- induction of apoptosis by mechanisms associated with microtubule dynamics;
- effect on the endothelium;
- mitochondrial dysfunction;
- proteolytic stress;
- changes in the functioning of the glutamatergic system in the central nervous system (CNS).

### 3.1. Disruption of Microtubule Dynamics and Axon Transport

The disruption of axon transport under chemotherapeutic agents is caused by a change in the microtubular architectonics of axons through inhibition (vinca alkaloids) or a sharp increase in the polymerization processes (proteasome inhibitors) of tubulin, the main microtubule protein that is involved in maintaining the cytoskeleton and intracellular transport of substances [26–30].

Microtubules are intracellular proteins that make up the cytoskeleton of eukaryotic cells, along with microfilaments and intermediate filaments. They are composed of two fundamental subunits, called α-tubulin and β-tubulin, bonded to each other to form polarized cylindrical structures in which α-tubulin is the negative terminal (−) and β-tubulin is the positive terminal (+). The third member of the tubulin family, γ-tubulin, plays a role in microtubule nucleation and assembly [30]. Microtubules polymerize and depolymerize continuously within the cell. This continuous assembly process occurs through the hydrolysis of GTP, which is bound to β-tubulin and is required for numerous vital cell functions, including cell division. During mitosis, microtubules assemble to form a mitotic spindle, which allows the reproduction and separation of chromosomes in two daughter cells [31]. Inhibition or enhancement of tubulin polymerization leads to a disruption in the structure of microtubules and, as a result, to a disruption in the processes of fast and slow axon transport, which consists in the movement of neurosecretory granules, glycoproteins, phospholipids, and some enzymes along the axon of a nerve cell, which are necessary for the synthesis of neurotransmitters in axon endings [32,33].

The pathogenesis of vinca alkaloid-induced CIPN is related to the mechanisms by which vincristine performs its antitumor function. Vincristine acts primarily by altering the normal function of microtubule assembly and disassembly, followed by blockage of mitosis and cell death. In addition to this, all other microtubule-related activities are inevitably compromised. Microtubules are found in abundance in neurons as they make up the skeleton of axons and dendrites, giving them their specialized morphology. By binding to microtubules, vincristine causes changes in the shape and stability of neurons, preventing retrograde and anterograde axonal transport with subsequent degeneration, which consists in the reabsorption of the distal nerve segment after its injury, and also causes a change in the transmission of nerve impulses and neuronal death [31]. Microtubules play a fundamental role in the myelination of nerve fibers as they are integral constituents of oligodendrocytes. Lee et al. (2020) demonstrated that vincristine destabilized microtubules, changing the structure and function of oligodendrocytes, followed by abnormal myelination and loss of peripheral sensory fibers. Although bortezomib is known to be a proteasome inhibitor, increased tubulin polymerization has been observed both in vivo and in vitro in CIPN models [32–34]. These changes in intracellular tubulin dynamics can lead to disruption of axonal transport.

### 3.2. Oxidative Stress

Some anticancer agents (platinum-based drugs, vinca alkaloids, proteasome inhibitors) enhance the production of reactive oxygen species (ROS) in cells. If the amount of ROS exceeds the capabilities of the antioxidant system and ROS accumulate in cells, then damage is initiated in it. This process is called oxidative stress [35]. This concept reflects excessive formation of free radicals and reactive oxygen species, which can damage proteins, nucleic acids, lipids through oxidation, which leads to adverse effects at all levels of organization of biological systems. Oxidative stress is a pathological condition and is observed in many diseases. But the CNS is particularly susceptible to oxidative stress, which is associated with a significant vulnerability of the nervous tissue itself due to its peculiarities [36]. CNS and peripheral nervous system has a high content of unsaturated lipids of variable valence metals (especially iron). Also of importance is the involvement of free radicals in neuroregulation and the ability of some neurotransmitters and hormones to promote formation of ROS. It is assumed that the onset of oxidative stress is associated with increased calcium transport and activation of calcium-dependent enzymes that damage

cell membranes, in particular gliocytes and neurons. Lipid peroxidation processes affect the integrity of axon membranes, and free-radical oxidation of neuronal proteins, including microtubules, impairs axonal transport. In addition, microglia, activated in the process of functioning, is a very powerful source of free radicals in the CNS and peripheral nervous system [35,36].

Neurotransmitter imbalance may also underlie oxidative stress. For example, as a catecholamine, dopamine is capable of autooxidation, which results in the formation of ROS as byproducts.It is based on lipid peroxidation, which primarily damages the membrane of neurons and their intracellular organelles (mitochondria, nuclei, lysosomes, endoplasmic reticulum). Oxidative stress plays an essential role in the development of the deficiency of nerve cell growth factors (neurotrophins) [36–38].

### 3.3. Apoptosis

Apoptosis is the genetically programmed death of neurons. Apoptosis induction is a pathogenetic mechanism of neurotoxicity characteristic of platinum preparations, taxanes, vinca alkaloids, and proteasome inhibitors. There are two pathways to trigger apoptosis: the receptor and mitochondrial pathway. The first receptor-mediated pathway is induced by physiological factors that induce apoptosis (cytokines, hormones, peptide growth factors) and begins with cell receptors specifically designed to turn on its program (Fas, TNFR1, DR3, DR4, DR5). These receptors in the extracellular region interact with inducers, specific ligands. It leads to the binding of their intracellular regions with adapters (cytoplasmic proteins) that facilitate the connection of receptors with inactive protease precursors from the caspase family, which are then activated as a result of proteolytic cleavage. The active caspases, in turn, affect mitochondria, which leads to the release of cytochrome C from them into the cytoplasm. The mitochondrial pathway of apoptosis activation is induced by DNA damage. It is associated with the activation of the p53 protein and the expression of genes encoding proteins of the Bcl-2, Bax, and Bid families. Causing a change in the permeability of the mitochondrial membrane, these proteins promote the release of cytochrome C [29]. The use of drugs that affect the dynamics of microtubules (eribulin, taxanes, vinca alkaloids and bortezomib) blocks the formation of the mitotic spindle and the cell passes into apoptosis with activation of p53 or by other mechanisms.

### 3.4. Mitochondrial Damage

Damage to mitochondria is one of the mechanisms of CIPN. Mitochondria are involved in the regulation of neurogenesis—neuronal differentiation, neuroplasticity formation, axon and dendrite formation [38]. Mitochondria also regulate neurotransmitter release through ATP synthesis and calcium deposition. It has been shown that vincristine can influence the movement of $Ca^{2+}$ across the mitochondrial membrane, decreasing both the amount and rate of $Ca^{2+}$ uptake and decreasing its efflux. Modification of mitochondrial absorption and $Ca^{2+}$ concentration alters mitochondrial function [39], which leads to increased neurotransmitter exocytosis [40] and ROS release [33]. These changes lead to a decrease in the excitability of neurons and glial function, activating apoptosis. This theory is supported by a large number of vacuolated mitochondria with disturbed cristae localized at the cell periphery [41–44].

### 3.5. Proteolytic Stress and Autophagy

The occurrence of intracellular reactions reflecting proteolytic stress is typical of bortezomib and is primarily manifested by the induction of heat shock proteins (HSPs) expression in cells [32]. Autophagy is a stress-induced cell survival program in which cells under metabolic, proteotoxic, or other stresses remove dysfunctional organelles and/or misfolded/polyubiquitinated proteins by shuttling them to the lysosome for degradation via specialised structures called autophagosomes. Oxaliplatin and taxanes, along with damage to the DNA of tumor cells, can induce their autophagy [45]. Due to their non-

specific action, it is possible that similar processes may also take place in the peripheral nervous system

### 3.6. Changes in the Functioning of the Glutamatergic System in the Central Nervous System

In addition to the peripheral nervous system, bortezomib can lead to dysregulation in the CNS. Since bortezomib cannot cross the blood-brain barrier, it probably causes CNS damage indirectly [46]. Elevated concentrations of glutamate, the largest neurotransmitter, were observed in the cerebrospinal fluid of rats treated with bortezomib [47]. In addition, bortezomib increased GFAP expression and altered astrocytic morphology in the spinal dorsal horn, while simultaneously decreasing the expression of the glutamate/aspartate transporter, an important extracellular glutamate uptake transporter expressed on astrocytes [48].

### 4. Protein-Protein Interactions on the Surface of EVs as Modulators of CIPN and the Effectiveness of Anticancer Therapy with Monoclonal Antibodies and Antibody-Drug Complexes (ADC)

Understanding of the pathogenesis of axonal degeneration of peripheral nerve fibers is based on the fact that peripheral axons and dorsal ganglia do not have a blood-brain barrier. This allows autoantibodies, immune complexes, toxic metabolites of cytostatics, and targeted drugs to directly contact nerve cells and nerve fibers and cause damage to myelin, tubulin, actin, and actin-binding proteins in them [24,25]. However, EVs can freely pass through the blood-brain barrier.

In terms of both the study of CIPN pathogenesis and feasibility of CIPN treatment, the study of the protein corona of the EVs, as well as intravesicular proteins, is very promising. In CIPN, the process of demyelination and damage to intraaxonal and neural proteins occurs in parallel with the regeneration of myelin via branching of dendrocytes and axons with the formation of new synapses that replace non-functioning ones [49]. In some cases, remyelination can be stimulated by antibodies produced by B cells. In this regard, an important role is given to antibodies that have protease activity and are capable of initiating myelin restoration [50]. Studies on antibodies with natural catalytic activity, which are called abzymes, are of particular interest. Recently, links between enzymatic activity of autoantibodies and neurodegenerative processes have been demonstrated in studies by A. A. Belogurov Jr. et al. (2008) and Doronin V.B. (2014) [50,51]. Abzymes with DNase, RNase, myelin hydrolyzing, and oxidoreductase activities have been found in the blood of patients with schizophrenia, multiple sclerosis, bronchial asthma, autoimmune thyroiditis, and systemic lupus erythematosus [50–55]. Taking into account the pathogenetic aspects of CIPN, antibodies with myelin hydrolyzing and oxidoreductase activities may be of the greatest importance in the pathogenesis of this variant of polyneuropathy.

The components of interstitial fluid and blood plasma are deposited on the surface EVs and form a protein corona, due to which EVs acquire a greater functional. Plasma-coated EVs had a higher density than nascent EVs and carried blood plasma proteins. Using confocal microscopy, capillary Western immunoassay, immune electron microscopy, and flow cytometry, E.A. Toth et al. (2021) identified nine common EVs corona proteins (ApoA1, ApoB, ApoC3, ApoE, complement factors 3 and 4B, fibrinogen α-chain, IgG2 and IgG4 heavy chains) that appear to be common corona proteins among EVs, viruses and artificial nanoparticles incubated with blood plasma [56].

Most patients with multiple myeloma are characterized by monoclonal gammopathy with overproduction of IgG antibodies. The association of monoclonal gammopathy with B-cell non-Hodgkin's lymphomas (NHL) is a well known phenomenon. Monoclonal gammopathy (MG) was detected in 17.2% of cases with B-cell NHL. IgG-MG was more frequent in cases with aggressive NHL, while IgM in cases with low risk NHL. MG was mostly associated with advanced stage and had not any prognostic significance on survival [57,58].

A high prevalence of autoimmune diseases was found in some types of cancer (lung cancer, multiple myeloma, NHL and Hodgkin's lymphoma), and according to different authors, it ranges from 5 to 12%, it increases significantly with age, a history of poly-

chemotherapy, radiation therapy, and targeted therapy [59,60]. In cancer patients, among a variety of neurological immune-mediated adverse events that occur against the background of the use of checkpoint inhibitors, autoimmune encephalitis, aseptic meningitis, Guillain-Barré syndrome, myasthenia gravis and myositis are of particular importance. The presence of autoimmune diseases associated with cancer or developed during antitumor therapy may be associated with both survival rates and the effectiveness of targeted therapy, quality of life, and serious side effects of the treatment, including polyneuropathy [61]. Literature data suggest the similarity of the mechanisms of formation of abzymes in blood plasma in lymphoproliferative and autoimmune diseases. Moreover, IgG with oxidoreductase activities have been shown in the blood of patients with breast and gastric cancer [62,63].

In addition to cancer cells, target antigens for therapeutic monoclonal antibodies and antibody-drug complexes (ADC) are also expressed on EVs. The antigen-presenting function of EVs is especially pronounced on salivary EVs, EVs derived from dendritic and immune cells. The list of antigens widely represented on circulating EVs includes HER2, CD20, CD30, CD33, BCMA, CD79b, PD-L1 [64,65]. Cases of EVs-mediated resistance to anticancer therapy with monoclonal antibodies and ADC have been described. For example, rituximab is an anti-CD20 mAb approved for the treatment of patients with CD20-positive NHL or chronic lymphocytic leukemia. Complement-dependent cytotoxicity plays a key role in the anticancer efficacy of rituximab. B-cell lymphoma cells can secrete CD20-bearing EVs, and rituximab binds to the surface of these EVs. Binding of rituximab results in complement fixation in EVs crown. This CD20+ EV-mediated decoy effect results in consumption of both complement and free rituximab, resulting in decreased efficacy of rituximab on target cancer cells [66]. Trastuzumab, an anti-HER2 monoclonal antibody approved for the treatment of patients with HER2-positive breast and stomach cancer. Trastuzumab also recruits immune effector cells that kill cancer cells through antibody-dependent cellular cytotoxicity. The HER2+ EV-mediated decoy effect may result in attenuation of the direct inhibitory effect of trastuzumab on tumor growth and the ADCC-related cytotoxic effect of trastuzumab [67].

In addition, the EVs protein crown is currently considered as a platform for complement activation. In the synovial fluid of patients with rheumatoid arthritis, significantly elevated level of leukocyte- derived EVs with complement components bound on their surface, including C1q, C4 and C3, were found. Red blood cell-derived EVs fix C1q followed by complement activation with binding of C3 fragments [68]. Several studies have shown that there is an association between the deposition of the membrane attack complex (MAC) in the retina and the progression of diabetic retinopathy. Complement activation by IgG-loaded EVs in blood plasma can lead to MAC deposition and contribute to endothelial damage and progression of diabetic retinopathy [69]. Similar data were obtained in the study of IgG-loaded EVs of astrocytic origin in the blood plasma of patients with Alzheimer's disease. It has been shown that circulating EVs from Alzheimer's disease patients can cause complement-mediated neurotoxicity, including the formation of a MAC. Astrocyte-derived EVs and, less effectively, neuronal-derived EVs from patients with Alzheimer's disease induced MAC expression on recipient neurons, membrane disruption, reduced neurite density, and decreased cell viability in rat cortical neurons and human iPSC-derived neurons. These effects were not induced by non-specific CD81+ EVs from blood plasma of Alzheimer's disease patients or by total blood plasma EVs from healthy donors [70]. Similar data are not available for Schwann cell-derived EVs. However, given the role of these vesicles in intercellular communication in the peripheral nervous system, a similar mechanism of complement-dependent neurotoxicity, which can be realized during antitumor therapy, can be assumed [37,71]. Protein-protein interactions on the surface of EVs as modulators of the effectiveness of anticancer therapy with monoclonal antibodies and antibody-drug complexes, as well as modulators of CIPN are presented in Figure 1.

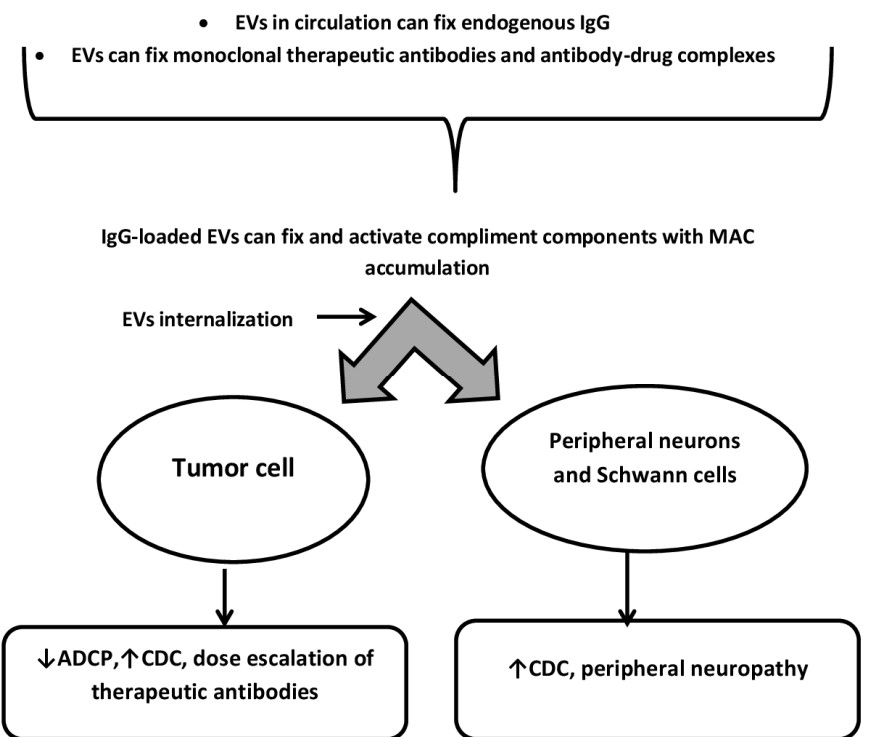

**Figure 1.** Protein-protein interactions on the surface of EVs as modulators of the effectiveness of anticancer therapy with monoclonal antibodies and antibody-drug complexes, as well as modulators of chemotherapy-induced polyneuropathy (CIPN). EVs in circulation can fix endogenous IgG, monoclonal therapeutic antibodies and antibody-drug complexes. Therefore, IgG-loaded EVs can fix and activate compliment components with membrane attack complex (MAC) accumulation. Thus, the use of targeted drugs can result in the immune-mediated cytotoxicity. mAb activation of the complement cascade lead to complement-dependent cytotoxicity (CDC) and complement receptor-mediated antibody-dependent cellular phagocytosis (ADCP). In the case of adhesion of IgG-loaded EVs on the surface of tumor cells or their internalization, enhancement of CDC and reduction in ADCP can occur. Internalization of vesicles with activated complement components by peripheral neurons and Schwann cells may also increase CDC and develop polyneuropathy.

The mechanisms of CIPN development that are not associated with IgG in the protein corona of exosomes are somehow related to the fact that many anticancer drugs induce apoptosis by different mechanisms. The expression of HSPs in cells is increased in response to the administration of drugs. HSPs in cells function as chaperones. They can be secreted into the blood and expressed in the exosome membrane. It is known that HSP27, being in the cell, effectively prevents polyneuropathy [72,73]. Moreover, bortezomib systemically inhibits the chymotrypsin-like and trypsin-like activity of the 20S and 26S proteasome pools in many cells. The manifestation of this process is especially critical for actively dividing cells [74]. In the context of neuropathy, not only neurons but also dividing neuroglial cells can be affected. Cells under these conditions experience proteolytic stress due to a large number of non-degraded proteins, and regulatory processes in the cell are disrupted. Given the large number of functions of neuroglial cells (supporting, regulatory, trophic, secretory, delimiting (Schwann cells), protective, neuronal learning function), it is of interest to study the role of glial cell-derived exosomes in the development of CIPN. In addition, the normal assembly of the 19S regulatory subunit of the 26S proteasome also requires the involvement of molecular chaperones [75]. We have previously found elevated levels of the 20S proteasome in ovarian and breast cancer tissues of the luminal subtype B, as well as in plasma exosomes of both breast and ovarian cancer patients compared with healthy women. The level of the 20S proteasome in plasma exosomes and ascites exosomes was

higher in ovarian cancer patients with low-volume ascites than in patients with moderate and high-volume ascites [76].

## 5. Internalization of Extracellular Vesicles by Neurons and Neuroglial Cells: Implications for the Treatment of CIPN

The internalization of EVs by neurons and neuroglia is the most critical moment for the therapeutic effect of EVs. It is known that the mechanisms of internalization of EVs can be determined both by protein-protein interactions on the cell surface and EVs, and by specific mechanisms of endocytosis and macropinocytosis [1,77]. In addition to internalization, the main variants of interaction of exosomes with the cell are the EVs adhesion on the cell membrane and the fusion of membranes of recipient cells and EVs membranes. The existence of process of fusion of vesicular membranes and the plasma membrane of the recipient cell, leading to the transfer of proteins anchored in the vesicular membrane without the transfer of intravesicular proteins and miRNA, is currently disputed [78]. The adhesion of exosomes to the cell membrane may reflect their antigen-presenting function. Exosomes produced by professional and nonprofessional antigen-presenting cells contain a large number of major histocompatibility complexes MHC, which suggests that exosomes have an antigen-presenting function [78]. In the context of the fact that the EVs protein corona itself can play a significant role in neuroprotection and taking into account that only a small part of the EVs is internalized by cells (no more than 15–20%), the study of EVs adhesion on the surface of neurons, axons, and neuroglia and regulation of this process is relevant in terms of the feasibility of CIPN therapy with EVs.

Extracellular vesicles derived from normal cells have previously been used to treat cancer and diabetic peripheral neuropathy in experimental models. For example, small extracellular vesicles or exosomes (sEVs) derived from fibroblasts carrying anti-Kras siRNAs have a therapeutic effect on a tumor in a mouse model of pancreatic cancer [79]. Exosomes derived from natural killer cells carrying high concentrations of perforin, granzyme B, and tumor necrosis factor to recipient cells in a mouse model have been successfully used to treat glioblastoma [80]. Intravenous administration of sEVs derived from healthy Schwann cells has been shown to markedly alleviate peripheral neuropathy in mice with diabetes [81].

Although agents targeting neuroprotection with anti-inflammatory and antioxidant properties have shown promising effects on the prevention and treatment of CIPN in patients and animal models, challenges remain in developing treatments for CIPN in which the therapy effectively inhibits CIPN without compromising the antitumor efficacy of chemotherapy drugs. Zhang et al. (2017) demonstrated that sEVs derived from mesenchymal stromal cells (MSC-sEVs) contain miRNAs and proteins that mediate neuronal function. MSC-sEVs can be locally internalized by the distal axons of cortical neurons and subsequently promote axonal growth even when their growth is inhibited [82]. The same authors showed that cerebral endothelial cell-derived sEVs (CEC-sEVs) in combination with a platinum preparation reliably suppressed CIPN and enhanced the antitumor effect of oxaliplatin in nude mice with human ovarian cancer. Intravenously injected CEC-sEVs were internalized by sciatic nerve fibers and cancer cells, and altered miRNA/protein networks that reduced neurotoxicity and enhanced the effects of platinum-based drugs. Having studied the full transcriptome profile of CEC-sEV miRNA, the authors found that exosomes were specifically enriched in miR-15b, -214, -125b. RT-PCR analysis showed that the level of these miRNAs in the nerve and tumor tissue was significantly reduced with the use of oxaliplatin. The authors attribute the therapeutic effect of CEC-sEVs to their internalization and transfer of the corresponding miRNAs [83].

## 6. Conclusions

Currently, the role of intravesicular proteins and the EVs protein corona in the development of CIPN is under research. It has been shown that EVs are effectively internalized by the mechanisms of endocytosis and macropinocytosis by neurocytes and glial cells, carry

markers of insulin resistance, functionally active proteins (receptors, cytokines, enzymes), and may be involved in the pathogenesis of drug-induced neuropathy. The mechanisms of CIPN associated with the EVs protein corona can be related with the accumulation of heavy chains of circulating IgG in it. G-class immunoglobulins in EVs are likely to have myelin hydrolyzing, superoxide dismutase, and oxidoreductase enzymatic activities. Moreover, circulating IgG-loaded EVs are a place for complement activation that can lead to membrane attack complex deposition in neuroglia and neurons. The mechanisms of CIPN development that are not associated with IgG in the EVs protein corona are somehow related to the fact that many anticancer drugs induce apoptosis of tumor cells, neurons, and neuroglial cells by various mechanisms. This process may be accompanied by the secretion of EVs with modified cargo (HSPs, 20S proteasomes, miRNAs).

**Author Contributions:** Conceptualization, N.V.Y. and L.P.S.; writing—original draft preparation, N.V.Y., N.O.P., I.N.U., T.S.K. and D.V.K.; writing—review and editing, N.V.Y. All authors have read and agreed to the published version of the manuscript.

**Funding:** This research did not receive any specific grant from funding agencies in the public, commercial, or not-for profit sectors.

**Institutional Review Board Statement:** Not applicable.

**Informed Consent Statement:** Unacceptably.

**Data Availability Statement:** All data presented in this study are available on request from the corresponding author.

**Conflicts of Interest:** The authors declare no conflict of interest.

## List of Abbreviations

CT—chemotherapy, CIPN—chemotherapy-induced polyneuropathy, EVs—extracellular vesicles, ROS—reactive oxygen species, CNS—central nervous system, MG—monoclonal gammopathy, HSPs—heat shock proteins, MSC-sEVs—small EVs derived from mesenchymal stromal cells, CEC-sEVs—cerebral endothelial cell-derived small EVs, NHL—Hodgkin's lymphomas, MAC—membrane attack complex, ADC—antibody-drug complexes, FOLFOX, XELOX, FLOX, FLOT—chemotherapy regimens.

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
