# Peer review of "The Role of Intravesicular Proteins and the Protein Corona of Extracellular Vesicles in the Development of Drug-Induced Polyneuropathy"

_cimb, doi:10.3390/cimb45040216_

Round 1

Reviewer 1 Report

Comments and Suggestions for Authors

This is a very well investigated and comprehensive review on the topic of CIPN. The authors have thoroughly explored and collected literature and have provided readers with a detailed coverage on the phenomenon.

Minor comments:

1.     Please include references for table 1.

2.     Correct typos in the text, ex. line 52 “neuropaties”., italicize “in vitro and in vivo” in line 145

3.     Is there any evidence of cancer/ tumors themselves increasing a patient’s vulnerability to peripheral neuropathy?

4.     The authors allude to EVs studied from patients with autoimmune/ other inflammatory conditions. Do these conditions also present with peripheral neuropathy as a secondary symptom/consequence of the primary disease? Are there are differences in the content of EVs in these patients vs. patients with cancers and/or patients with cancers receiving chemotherapy? Though it is briefly discussed, including this in the manuscript with little more elaboration, perhaps even a table/illustration if space would allow for it can provide significant insight in understanding disease pathogenesis of CIPN (that has been otherwise very comprehensively covered by the authors).

5.     Finally, the authors have cited appropriate works for all their text. However, it appears that large portions of the writing are often supported by references that are inserted at the very end of the portion of the text, which makes it appear like some lines of text do not have reference. Can the authors include references (even if repetitive) in appropriate positions of the text, so readers can refer to the literature, if they choose to do so for a topic/statement of interest?

Author Response

  1. Please include references for table 1.

Answer: references added to table 1

  1. Correct typos in the text, ex. line 52 “neuropaties”., italicize “in vitro and in vivo” in line 145.

Answer: grammatical errors corrected

  1. Is there any evidence of cancer/ tumors themselves increasing a patient’s vulnerability to peripheral neuropathy?

Answer: There is no consistent evidence in the literature on whether the presence of a malignancy increases the risk of developing peripheral polyneuropathy. However, in the first version of the review, in the subchapter “Chemotherapy-induced polyneuropathy as a clinical challenge”, the authors note that “The risk of damage to the peripheral nervous system is increased in patients with hereditary and acquired neuropathies; suffering from diabetes; alcohol abusers; having hepatic and/or renal dysfunction; previously treated with any neurotoxic drugs (especially vinca alkaloids, taxanes, platinum derivatives, methotrexate) [13]. In the new version of the review, this phrase is corrected to read “The risk of damage to the peripheral nervous system is increased in cancer patients with hereditary and acquired neuropathies; suffering from diabetes; alcohol abusers; having hepatic and/or renal dysfunction; previously treated with any neurotoxic drugs (especially vinca alkaloids, taxanes, platinum derivatives, methotrexate) [13]”.

  1. The authors allude to EVs studied from patients with autoimmune/ other inflammatory conditions. Do these conditions also present with peripheral neuropathy as a secondary symptom/consequence of the primary disease? Are there are differences in the content of EVs in these patients vs. patients with cancers and/or patients with cancers receiving chemotherapy? Though it is briefly discussed, including this in the manuscript with little more elaboration, perhaps even a table/illustration if space would allow for it can provide significant insight in understanding disease pathogenesis of CIPN (that has been otherwise very comprehensively covered by the authors).

Answer: Many autoimmune diseases, such as multiple sclerosis, Guillain-Barré syndrome, chronic inflammatory demyelinating polyneuropathy, are accompanied by symptoms of peripheral polyneuropathy. The authors did not delve into this problem, and wanted to focus on the role of extracellular vesicles in the development of polyneuropathy induced by anticancer drugs. We believe that extracellular vesicles are also involved in the development of polyneuropathy caused by an autoimmune disease, but how is a topic for a separate review.

  1. Finally, the authors have cited appropriate works for all their text. However, it appears that large portions of the writing are often supported by references that are inserted at the very end of the portion of the text, which makes it appear like some lines of text do not have reference. Can the authors include references (even if repetitive) in appropriate positions of the text, so readers can refer to the literature, if they choose to do so for a topic/statement of interest?

Answer: The authors worked to ensure that the references in the review were cited in a timely manner, the list of references was checked and edited.

Reviewer 2 Report

Comments and Suggestions for Authors

The author's manuscript is written in clear scientific language. It is devoted to a review of studies in the field of the role of intravesicular proteins and the protein crown of extracellular vesicles in the development of polyneuropathy induced by anticancer drugs.

There are a few notes on the manuscript:

1) At the beginning of the review, the authors talk about the important role of extracellular vesicles in the pathogenesis of diabetic polyneuropathy, but this position is not developed further. Additional chapter needed.

2) Figures 1 and 2 somewhat duplicate each other. Suggestion - to present figure 2 in the form of a graphic abstract.

3) Minor linguistic edit required throughout the review test

Author Response

  • At the beginning of the review, the authors talk about the important role of extracellular vesicles in the pathogenesis of diabetic polyneuropathy, but this position is not developed further. Additional chapter needed.

Answer. A section “Proteins and miRNAs of extracellular vesicles (EVs) as biomarkers of diabetic polyneuropathy” was added to the review, which presents literature data on proteins and miRNAs of extracellular vesicles as markers of diabetic polyneuropathy, nephropathy, and endothelial dysfunction. References have been added to the reference list.

2) Figures 1 and 2 somewhat duplicate each other. Suggestion - to present figure 2 in the form of a graphic abstract.

 Answer: Careful work has been done on the structure and style of the review. At the moment the review contains 6 sections, 1 table and 1 figure. Figure 2 has been removed since it almost completely duplicates the graphic abstract.

3) Minor linguistic edit required throughout the review test

Answer: The English-language correction was carried out in the translation service of the Cancer Research Institute of the Tomsk National Research Medical Center of the Russian Academy of Sciences. We are ready to use the language service at MDPI after the final decision of the editor, taking into account the changes already made to the text of the manuscript after the review.

Reviewer 3 Report

Comments and Suggestions for Authors The manuscript contains an overview of modern ideas about the role
of
intravesicular proteins and the protein corona of  extracellular
vesicles
in the development of chemotherapy-induced polyneuropathy 
The molecular mechanisms underlying these effects are considered.
The article may be useful for specialists developing anti-cancer drugs,
as well as clinicians working with real patients
.

Author Response

Answer. The authors thank the referee for the high evaluation of the manuscript.